# Does postoperative plasma IL-6 improve early prediction of infection after pulmonary cancer surgery? A two-centre prospective study

Ted Reniers[1,2]*, Peter G. Noordzij[1,2], Eelco J. Veen[3], Erik F.N. Hofman[4],
Anne Marlies Taselaar[5], W. Anton Visser[5], Pim van der Heiden[5], Stefan Boeckx[5],
Judith M.A. Emmen[6], Ineke M. Dijkstra[7], Olaf L. Cremer[2], Lisette M. Vernooij[1,2],
Thijs C.D. Rettig[5]

1 Department of Anaesthesiology, Intensive Care and Pain Medicine, St Antonius Hospital, Nieuwegein, Zuid-Holland, the Netherlands, 2 Department of Anaesthesiology and Intensive Care, University Medical Centre Utrecht, Utrecht, the Netherlands, 3 Department of Surgery, Amphia Hospital, Breda, Noord-Brabant, the Netherlands, 4 Department of Surgery, St Antonius Hospital, Nieuwegein, Zuid-Holland, the Netherlands, 5 Department of Anaesthesiology and Intensive Care, Amphia Hospital, Breda, Noord-Brabant, the Netherlands, 6 Result Laboratory for Clinical Chemistry and Haematology, Amphia Hospital, Breda, Noord-Brabant, the Netherlands, 7 Department of Clinical Chemistry, St Antonius Hospital, Nieuwegein, Zuid-Holland, the Netherlands

* t.reniers@antoniusziekenhuis.nl

## Abstract

### Introduction

Postoperative hyperinflammation increases infection risk. We hypothesized that interleukin-6 (IL-6) is an early predictor of infection after pulmonary cancer surgery.

### Methods

A two-centre prospective cohort study, including consecutive elective pulmonary cancer surgery patients. The primary outcome was any postoperative infection within 30 days. Multivariable logistic regression was used to create a core model (age, sex, surgery duration and Charlson comorbidity index) to which maximum IL-6, C-reactive protein (CRP), procalcitonin (PCT) concentrations and white blood cell count (WBC) between start of anaesthesia and 24 hours were added. The predictive performance of the models was assessed.

### Results

170 patients were analysed, of whom 38 (22%) developed a postoperative infection. IL-6 concentrations peaked 6 hours postoperatively, whereas CRP had not yet reached peak levels at 24 hours (time of prediction). Maximum IL-6 concentrations were associated with postoperative infection (adjusted odds ratio (aOR) 1.04 per 10 pg/ml, 95% confidence interval (CI) 1.00–1.09, p=0.047) as was CRP (aOR 1.01 per mg/L, 1.00–1.03, p=0.032). WBC and PCT were not associated with postoperative infection. The

**Data availability statement:** The data is available from https://doi.org/10.17605/OSF.IO/MSJ65.

**Funding:** Funding for biomarker analysis was provided by Roche Diagnostics. Roche Diagnostics had no role in the design and conduct of the study.

**Competing interests:** P.G. Noordzij is a member of the advisory board of Roche Diagnostics on the perioperative use of biomarkers. P.G. Noordzij, T.C.D. Rettig and T. Reniers conduct a separate research study on perioperative biomarkers funded by Roche Diagnostics. This does not alter our adherence to PLOS ONE policies on sharing data and materials. All other authors have no conflicts of interest.

c-statistic of the prediction models that included IL-6 or CRP concentrations were 0.67 (95%CI: 0.56–0.77) and 0.68 (0.57–0.77), respectively, compared to 0.67 (0.56–0.76) for the core model. IL-6 and CRP slightly improved calibration by broadening the range of predicted probabilities. Reclassification did not improve.

## Conclusion

Plasma IL-6 and CRP levels observed within 24 hours from the start of surgery are associated with postoperative infection risk, yet the added value of these biomarkers to a simple clinical prediction model seems limited.

## Introduction

The incidence of postoperative infection after pulmonary cancer surgery is approximately 20% [1–3]. Postoperative infection can progress to sepsis, defined as life-threatening organ dysfunction caused by a dysregulated host response to infection [4]. In roughly one-third of cases, sepsis is fatal and those who survive sepsis are at increased risk for persistent functional disability [5,6]. Timely initiation of antibiotics is crucial to prevent organ failure and mortality [7]. Early identification of patients who are at increased risk for postoperative infection could enhance surveillance for infectious signs and symptoms and may ensure prompt diagnosis and treatment.

Postoperative hyperinflammation might be an important predictor to identify these patients at risk for infection. Damage associated molecular patterns (DAMPs), released after surgical trauma, activate the innate immune system resulting in both a pro- and anti-inflammatory response to promote tissue healing and protect the body from invading pathogens. In some patients, the inflammatory response to surgery is excessive, resulting in hyperinflammation which increases the risk of postoperative infection [8,9]. Both interleukin-6 (IL-6) and C-reactive protein (CRP) are inflammatory biomarkers that increase dependent on injury severity [10]. Compared to CRP, IL-6 has more rapid dynamics. IL-6 reaches peak concentrations within hours after surgery and has a short plasma half-life, whereas CRP concentrations peak up to three days after surgery [11,12]. Therefore, IL-6 could be a more suitable biomarker to assess a patient's risk of infection on the first postoperative morning, leaving a clinically relevant time window for infection risk mitigation and surveillance. Currently, literature considering IL-6 as a predictor for postoperative infection in pulmonary surgery is scarce [11].

Our primary aim was to investigate postoperative hyperinflammation, assessed by maximum IL-6 concentrations, as an early predictor for infection after pulmonary cancer surgery. Secondarily, the incremental predictive performance of IL-6, CRP, white blood cell count (WBC) and procalcitonin (PCT) to a core model for predicting infection was evaluated.

## Materials and methods

### Design and participants

This prospective observational two-centre cohort study was conducted in the Amphia Hospital Breda and the Sint Antonius Hospital Nieuwegein, the Netherlands. Patients

were recruited between September 3rdh 2018 and April 29th 2022. Patients undergoing elective pulmonary surgery (pneumonectomy, (bi)(sleeve)lobectomy, segmentectomy) for cancer with an American Society of Anesthesiologists (ASA) physical status classification of ≥2 and a planned postoperative admission to the Intensive Care Unit (ICU) were eligible for inclusion. Patients with a suspected infection at the time of surgery or those who required reoperation within 24 hours of surgery were excluded. The study was preregistered on the 13th of July 2018, before conducting the research with The Netherlands Trial Register (NTR), currently available via the International Clinical Trial Registry Platform (ICTRP), with registration number NL-OMON25075. Ethical approval of the study protocol was obtained on the 23rd of July 2018 by the 'Toetsingscommissie Wetenschappelijk Onderzoek Rotterdam e.o.' (TWOR, number NL.64754.101.18).

All patients provided written informed consent. The study was conducted by the principles of the Declaration of Helsinki. This study was reported according to the 'Transparent reporting of a multivariable prediction model for individual prognosis or diagnosis (TRIPOD): The Tripod statement' [13].

## Study procedures and biomarker sampling

All patients underwent surgery under general anaesthesia or general anaesthesia combined with epidural analgesia or erector spinae plane block. Patients received a single dose of two grams of cefazolin intravenous (IV) prophylaxis before surgery and 8 mg IV dexamethasone after anaesthesia induction as postoperative nausea and vomiting (PONV) prophylaxis, unless contraindicated. The surgical approach was primarily minimally invasive by video-assisted thoracoscopic surgery (VATS) or robot-assisted thoracoscopic surgery (RATS). Perioperative aesthetic management was at the discretion of the anaesthesiologist and all patients were postoperatively admitted to the Intensive Care Unit (ICU) according to hospital protocols.

Blood samples were drawn from an arterial line directly after induction of general anaesthesia (further referred to as 'preoperative sample'), and after 6, 9, 12, 24, 48, and 72 hours. At each time point, a volume of 12 ml blood was drawn in three lithium heparin aliquots and one K3EDTA aliquot. Plasma samples were centrifuged and stored in a freezer at −80°C and shipped to the clinical chemistry lab of the Sint Antonius Hospital Nieuwegein for batch analysis. Measurements of plasma IL-6, CRP, and PCT were performed on an automated Cobas® 8000 platform (Roche Diagnostics, Mannheim, Germany). WBC were measured in fresh blood samples using the XN-9000 platform (Sysmex Corporation, Kobe, Japan).

## Primary outcome

The primary outcome was the occurrence of postoperative infection within 30 days of surgery, including respiratory infection, superficial-, deep- and organ/ space surgical site infection (including empyema), urinary tract infection or sepsis. Sepsis was defined as a quick SOFA score of ≥2 in response to an infection. Infections were defined according to the Centers for Disease Control and Prevention (CDC) criteria (S1 File). Outcome assessment was done by an intensive care physician together with a thoracic surgeon. If they did not achieve a consensus, another intensive care physician was consulted as a third assessor. All assessors were blinded for the study biomarker results. Standard care biomarker results (e.g., CRP and WBC), as ordered by the treating physician, were available during outcome assessment.

## Core model for infection

We aimed to assess the (incremental) predictive value of maximum inflammatory biomarker concentrations within 24 hours from the start of surgery, relative to a core model. Since, to the best of our knowledge, no generally used postoperative infection prediction tool exists for patients undergoing pulmonary surgery, we developed a core model using widely available clinical parameters. The core model existed of the variables age, sex, surgery duration (as a proxy for the extent of surgery) and the Charlson Comorbidity Index (CCI) (reflecting overall patient comorbidities) based on clinical reasoning [14].

## Sample size calculation

The sample size was estimated according to the rule of thumb of 10 events per predictor variable in the multivariable model. With an expected postoperative infection incidence of approximately 20%, a sample size of 250 patients was set to test five predictor variables [1,2]. At study design, postoperative admission to the intensive care unit (ICU) was the standard of care. During the study period, however, 'rapid recovery' pathways after pulmonary surgery were introduced. Consequently, patients were increasingly discharged to the general ward and, in patients that were admitted to the ICU, arterial lines were removed as soon as possible. Considering the frequency of blood sampling in the study design, the burden of blood sampling without an arterial line would be too high and conflicting with the study procedures as approved by the medical ethics committee. Additionally, study recruitment was lagging due to competing studies and the COVID-19 pandemic. Therefore, patient recruitment was ended before the desired sample size was met.

## Missing data

Missing data was apparent for IL-6 (8%), CRP (8%), PCT (8%) and WBC (6%). Data is often not missing completely at random, so complete case analysis can be biased [15]. Therefore, missing data was imputed in 15 imputation sets using the multiple imputation by chained equation (MICE) technique.

## Statistical analysis

Complete case analysis was used to present baseline characteristics and descriptive data of perioperative biomarker concentrations. Baseline characteristics were presented as the median and interquartile range (IQR) for continuous data and number and percentage for categorical data. To investigate differences in perioperative IL-6, CRP, WBC and PCT concentrations between patients with infection and without infection, median values (IQR) were plotted for each time point. The Mann-Whitney test was used to analyse differences in continuous data and the Chi-square test for categorical data between patients with or without postoperative infection.

We used uni- and multivariable logistic regression analysis to investigate the association between postoperative maximum biomarker concentrations within 24 hours from the start of surgery and postoperative infection. To assess incremental predictive performance, a core model was constructed and thereafter, maximum biomarker concentrations were added. The incremental predictive performance was only analysed if the addition of the biomarker significantly improved model fit according to the likelihood ratio test (LRT, $p < 0.05$). Discrimination was assessed using the c-statistic and calibration by a calibration plot. Results from logistic regression analysis and predictive performance measures were pooled using Rubin's Rule. We used a reclassification table to assess the reclassification of infection risk after the addition of the biomarkers to the core model. To this end, we calculated the predicted infection risk for each individual in the original, non-imputed dataset using the estimates of the pooled regression models after MICE. Additionally, we calculated the net reclassification index (NRI) for which a cut-off for elevated infection risk of >20% was used. p-value of <0.05 was considered statistically significant. All analyses were performed using R version 4.3.1.

## Results

### Study population

193 patients were included in this study. Five patients were excluded because they did not have a malignancy in retrospect, in five patients no resection was performed, and two patients used preoperative antibiotics for suspected infection. In 11 patients, no biomarker samples were available at any of the study time points due to logistical issues. Hence, 170 patients were included in the analysis (S1 File).

The median age was 67 years (IQR 60,73) and 76 (45%) patients were female. The CCI was ≥ 3 in 30 (18%) patients. In the majority of patients, the surgical approach was minimally invasive by VATS or RATS (N = 137, 81%) and 140 (83%)

patients underwent a lobectomy or sleeve resection. Median surgery duration was 193 minutes (165,227) and 151 (89%) patients received dexamethasone as PONV prophylaxis (Table 1).

## Postoperative infection

In total, 38 (22%) patients developed a postoperative infection, with a total of 45 infection events (i.e., some patients developed more than one infection event). 28 (74%) infection events were respiratory related, seven (18%) patients had a surgical site infection and three (8%) had a urinary tract infection. Seven (18%) patients developed postoperative sepsis. One patient had a postoperative infection on the first day after surgery. The median timing of infection diagnosis was four days after surgery, and 66% of infections were diagnosed in the first postoperative week. Patients with infection had longer surgery duration, more intraoperative blood loss and less frequently received IV dexamethasone.

**Table 1. Baseline Characteristics of patients with and without postoperative infection.**

| Characteristics | Infection (N = 38) | No infection (N = 132) | p-value |
|---|---|---|---|
| Age (years) | 68 (61,73) | 67 (60,73) | 0.964 |
| Male sex | 23 (61) | 71 (54) | 0.582 |
| BMI (Kg/m$^2$) | 24.4 (22.5,27.4) | 25.6 (23.6,28.9) | 0.023 |
| Active smoker | 11 (28.9) | 37 (28.0) | 1.000 |
| COPD | 24 (63.2) | 53 (40.2) | 0.020 |
| Cardiovascular Disease | 12 (31.6) | 41 (31.1) | 1.000 |
| Chronic Kidney Disease | 1 (2.6) | 11 (8.3) | 0.395 |
| Diabetes | 1 (2.6) | 22 (16.7) | 0.050 |
| ASA class ≥3 | 19 (50.0) | 58 (43.9) | 0.634 |
| Charlson Comorbidity Index ≥3 | 6 (15.8) | 24 (18.2) | 0.921 |
| ECOG performance score | | | 0.422 |
| 0 | 26 (68.4) | 100 (75.8) | |
| 1 | 12 (31.6) | 30 (22.7) | |
| ≥2 | 0 (0.0) | 2 (1.5) | |
| Preoperative Methotrexate | 3 (7.9) | 3 (2.3) | 0.227 |
| Preoperative Steroids | 2 (5.3) | 3 (2.3) | 0.677 |
| TNM class | | | 0.073 |
| 1 | 15 (44.1) | 68 (56.7) | |
| 2 | 8 (23.5) | 34 (28.3) | |
| ≥3 | 11 (32.4) | 18 (15.0) | |
| Surgery Type | | | 0.768 |
| Segment/wedge | 6 (15.8) | 16 (12.2) | |
| (Bi)lobectomy/ sleeve | 30 (78.9) | 110 (84.0) | |
| Pneumonectomy | 2 (5.3) | 5 (3.8) | |
| Open surgery | 12 (31.6) | 21 (15.9) | 0.055 |
| Surgery duration in minutes | 223 (181,260) | 186 (158,222) | 0.002 |
| Epidural analgesia | 26 (68.4) | 87 (65.9) | 0.925 |
| Intraoperative steroids | 30 (78.9) | 121 (91.7) | 0.057 |
| Intraoperative blood loss (ml) | 300 (100,475) | 150 (50,250) | 0.003 |

Continuous data is presented in median (IQR) categorical data as count (%). BMI = body mass Index, COPD = chronic obstructive pulmonary disease, cardiovascular disease = a composite of stroke, heart failure, coronary artery disease, peripheral artery disease, ASA = American Society of Anesthesiologists, ECOG = Eastern Cooperative Oncology group, TNM = classification of malignant tumors. Missing data count (%): Surgery type = 1 (0.6%), preoperative methotrexate = 1 (0,6%), TNM class = 16 (9.4%).

## Biomarker kinetics

Preoperative median IL-6 and CRP concentrations were higher in patients with postoperative infection compared to non-infected patients (6.1 pg/ml (IQR 3.3,10.9) vs. 3.4 pg/ml (2.0,6.0), p = 0.002 and 3.4 mg/L (2.1,22.1) vs. 1.8 mg/L (0.8,6.0), p = 0.016 for IL-6 and CRP respectively, Fig 1A, B). Postoperative peak IL-6 concentrations occurred after six hours in the majority of patients (N = 143, 70%), while maximum CRP concentrations were reached on postoperative day three in the majority of patients (N = 84, 50%). Median maximum concentrations of both IL-6 and CRP within 24 hours from the start of surgery were higher in patients with postoperative infection (135.0 pg/ml (IQR 80.0,180.5) vs. 80.5 pg/ml (54.8,131.5), p = 0.003 and 58.4 mg/L (38.2,88.6) vs. 44.4 mg/L (29.5,60.2), p = 0.009 for IL-6 and CRP respectively (S1 File). Such differences were not observed for WBC and PCT, across all time points (Fig 1C, D).

## Predictive performance

IL-6 was associated with postoperative infection, independent from the core model predictors (aOR 1.04 per 10 pg/ml, (95% CI 1.00,1.09)) (Table 2), and improved model fit when added to the core model (LRT p = 0.048). The c-statistic of

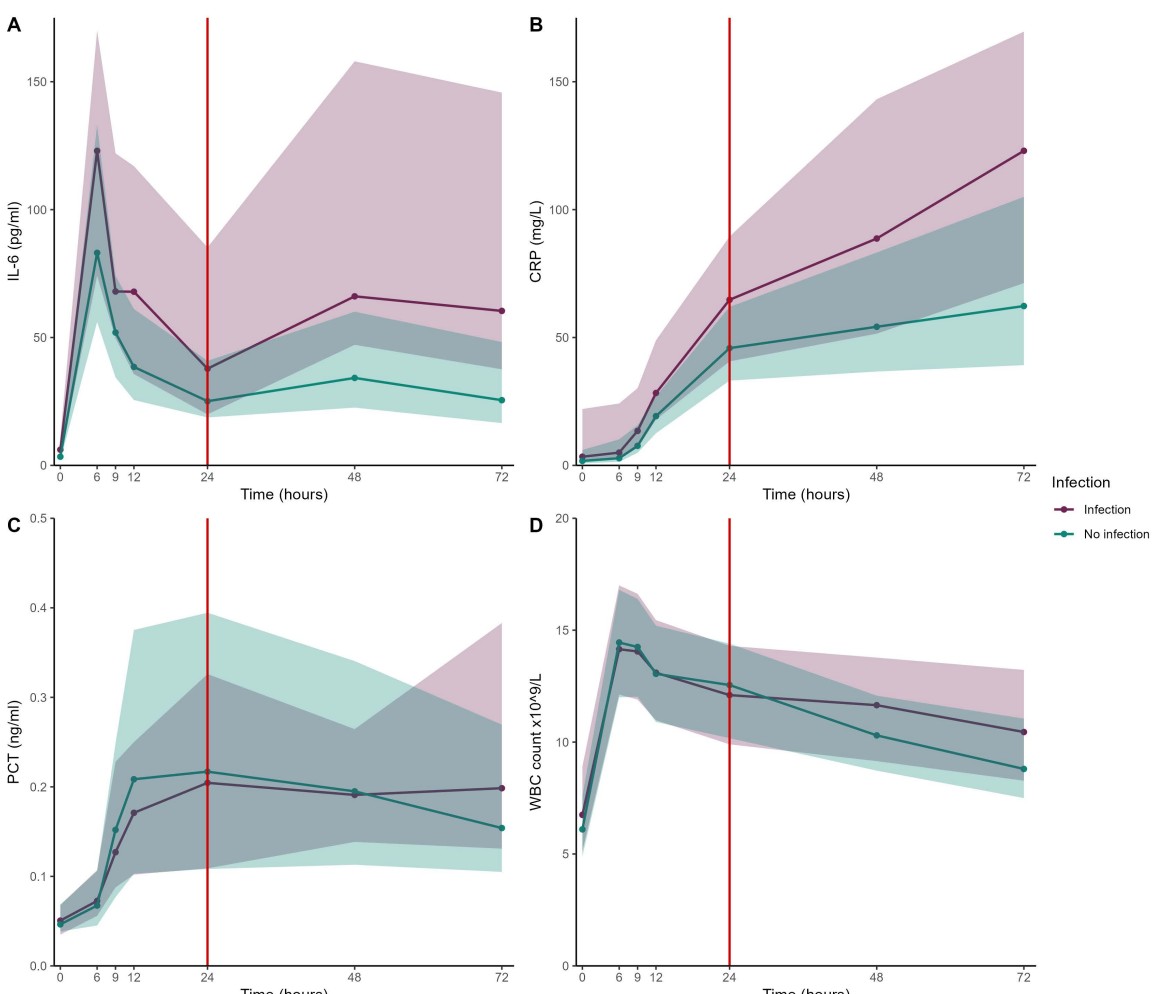

**Fig 1. Median perioperative biomarker concentrations.** Median (interquartile range) concentrations of: A interleukin 6 (IL-6), B C-reactive protein (CRP), C procalcitonin (PCT) and D White blood cell count (WBC) between the start of surgery (T = 0 hours), time of prediction (T = 24 hours, red vertical line) and 72 hours postoperatively for patients with and without a postoperative infection.

**Table 2. Multivariable logistic regression analyses on 30-day postoperative infection.**

| Multivariable Analysis | | | |
|---|---|---|---|
| Predictor | OR (95% CI) | C-statistic (95% CI) | p-value |
| **Core model** | | 0.67 (0.56, 0.76) | |
| Age (years) | 1.01 (0.96,1.05) | | 0.814 |
| Male sex | 1.16 (0.53,2.55) | | 0.711 |
| Surgery duration (mins) | 1.01 (1.00,1.02) | | 0.010 |
| CCI ≥ 3 | 0.69 (0.25,1.96) | | 0.488 |
| **Core model + IL-6 per 10 pg/ml** | | 0.67 (0.56, 0.77) | |
| Age (years) | 1.00 (0.96,1.05) | | 0.898 |
| Male sex | 1.23 (0.55,2.77) | | 0.616 |
| Surgery duration (mins) | 1.01 (1.00,1.01) | | 0.079 |
| CCI ≥ 3 | 0.66 (0.23,1.91) | | 0.445 |
| Max IL-6 at T24 | 1.04 (1.00,1.09) | | 0.047 |
| **Core model + CRP per mg/L** | | 0.68 (0.57, 0.77) | |
| Age (years) | 1.00 (0.96,1.05) | | 0.897 |
| Male sex | 1.09 (0.49,2.43) | | 0.829 |
| Surgery duration (mins) | 1.01 (1.00,1.01) | | 0.116 |
| CCI ≥ 3 | 0.66 (0.23,1.94) | | 0.451 |
| Max CRP at T24 | 1.01 (1.00,1.03) | | 0.032 |
| **Core model + PCT per ng/ml** | | NA | |
| Age (years) | 1.00 (0.96,1.05) | | 0.830 |
| Male sex | 1.10 (0.50,2.44) | | 0.808 |
| Surgery duration (mins) | 1.01 (1.00,1.02) | | 0.008 |
| CCI ≥ 3 | 0.69 (0.24,1.96) | | 0.481 |
| Max PCT at T24 | 1.33 (0.92,1.92) | | 0.131 |
| **Core model + WBC*10^9/L** | | NA | |
| Age (years) | 1.01 (0.96,1.05) | | 0.753 |
| Male sex | 1.15 (0.52,2.54) | | 0.723 |
| Surgery duration (mins) | 1.01 (1.00,1.02) | | 0.011 |
| CCI ≥ 3 | 0.71 (0.25,2.00) | | 0.513 |
| Max WBC at T24 | 1.02 (0.93,1.11) | | 0.697 |

Odds ratios (OR) and concordance statistic (c-statistic) with 95% Confidence Intervals (CI). IL-6 = Interleukin 6, CRP = C-reactive protein, PCT = procalcitonin, WBC = white blood cell count, CCI = Charlson Comorbidity Index, T24 = 24 hours after start surgery.

the core model was 0.67 (95% CI 0.56,0.76) and 0.67 (0.56,0.77) when IL-6 was added. The calibration plot of the model including IL-6 showed a slightly broader range of predicted probabilities (Fig 2B). Maximum CRP concentrations were also associated with postoperative infection (aOR 1.01, (95% CI 1.00,1.03)) and significantly improved model fit (LRT p = 0.032). Discrimination of the CRP model was similar to the IL-6 model (c-statistic 0.68 (95% CI 0.57,0.77), as was an improvement in calibration, with a slightly increased range in predicted values (Fig 2C). Adding both maximum CRP and maximum IL-6 concentrations to the core model did not further improve model fit (LRT p = 0.218). Neither PCT nor WBC improved model fit (LRT p > 0.05). Therefore, we did not further assess the predictive performance of these biomarkers.

### Reclassification of infection risk

Fig 3 shows the reclassification of patients for postoperative infection. In one patient reclassification was not assessed since biomarker results were missing. Adding IL-6 to the core model did not improve the classification of infected

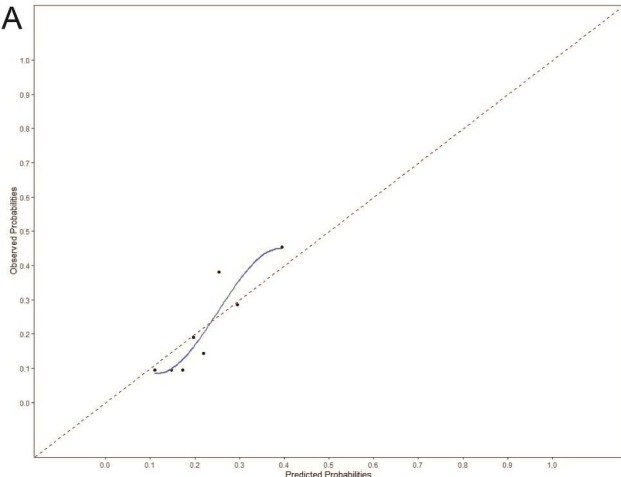

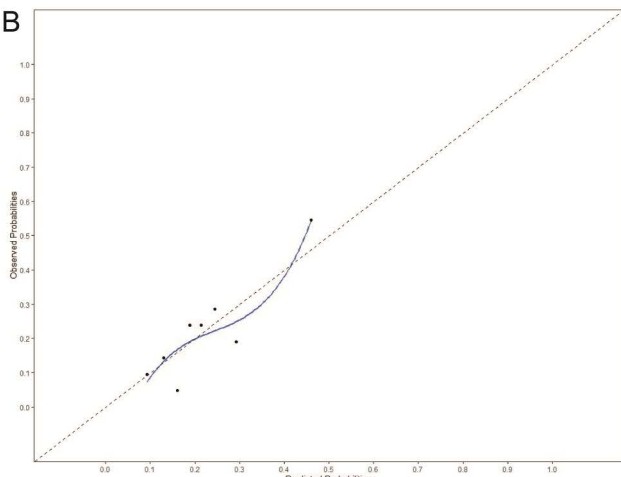

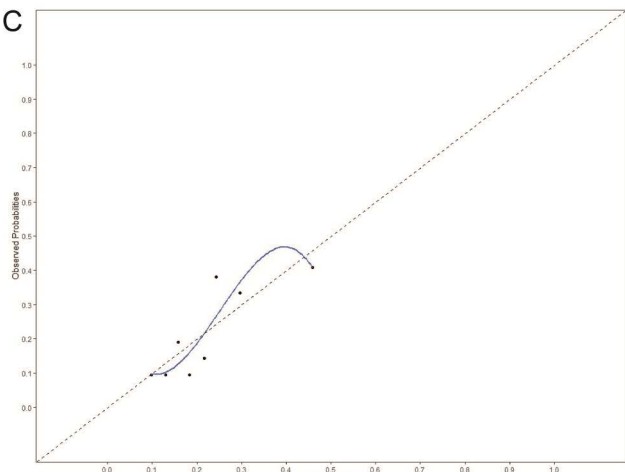

**Fig 2. Calibration plots.** Calibration plots of A the core model, B the core model with IL-6 added, and C the core model with CRP added. Each dot in the figure represents 1/8th of the total study population.

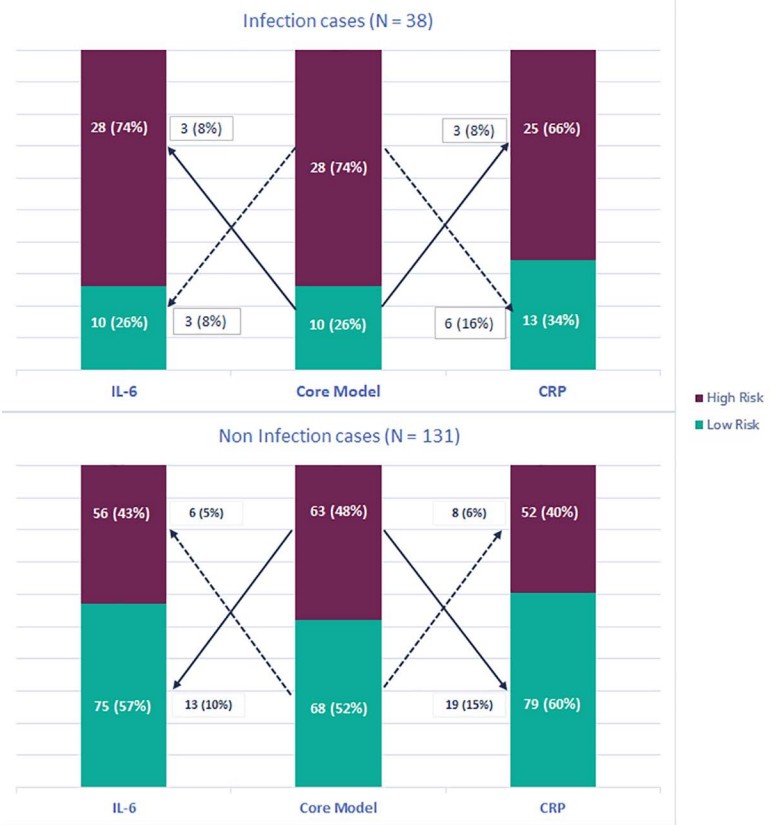

**Fig 3. Infection risk reclassification.** A cut-off for elevated infection risk of >20% was used. Solid arrow: correct reclassification, dashed arrow: incorrect reclassification. IL-6 = Interleukin 6, CRP = C-reactive protein.

patients, whereas CRP worsened the classification of infected patients. IL-6 and CRP both improved the classification of non-infected patients. The NRI was 0.05 (95% CI −0.08, 0.19) and 0.005 (−0.16,0.18) for IL-6 and CRP respectively.

## Discussion

This study analysed postoperative concentrations of inflammatory biomarkers to improve early prediction of infection after pulmonary cancer surgery. Maximum IL-6 and CRP concentrations within 24 hours from the start of surgery were associated with postoperative infection. In most patients, IL-6 peaked 6 hours postoperatively whereas CRP concentrations were still rising 72 hours postoperatively. Based on the predictive performance measures discrimination, calibration and reclassification, the added value of IL-6 and CRP to a simple clinical prediction model seems limited. Maximum postoperative WBC and PCT were not associated with postoperative infection.

Our study builds upon previous research in which the authors concluded univariable peak plasma IL-6 concentrations were associated with severe postoperative complications after pulmonary surgery [11]. We observed similar results regarding infection, although further exploration of predictive performance measures in a multivariable approach suggested limited clinical relevance of IL-6.

Several other studies investigated the association between inflammatory biomarkers and postoperative infection in various surgery types, however, most of these studies used an univariable, diagnostic approach [9,16]. In daily clinical practice, inflammatory biomarkers are routinely assessed, primarily to diagnose infection. For diagnostic purposes,

a biomarker is used to determine whether there is an infection at the time of biomarker assessment (i.e., the disease is present at the time of the diagnostic test) and, if so, the disease can be treated. When predicting postoperative infection, an inflammatory biomarker is used to assess the risk of developing an infection in the future (i.e., the disease is not present at the time of biomarker assessment). Identifying patients at risk for postoperative infections could improve infection surveillance and guide preventative strategies. These measures can only be effective when the prediction horizon (i.e., the elapsed time between the moment of prediction and postoperative infection) is sufficiently large. So, prediction research warrants a different methodological approach compared to diagnostic research and both approaches have distinct clinical implications. In abdominal surgery, a meta-analysis evaluating CRP to predict postoperative infection stated that CRP performs best three to four days after surgery [9]. By this time, however, a substantial part of patients are already diagnosed with postoperative infection, making infection prediction futile. On postoperative day one, when a postoperative infection is unlikely to be present, the predictive performance of CRP as a single test in abdominal surgery was only moderate [9]. Since, in contrast to CRP, IL-6 already peaks within 24 hours from the start of surgery, we hypothesized that IL-6 would be superior to CRP in predicting postoperative infection on the first postoperative morning after pulmonary surgery.

Although IL-6 did not outperform CRP as an early predictor for postoperative infection in our study, it would be premature to conclude that the predictive performance of both biomarkers is equal. In a prior study, investigating IL-6 and CRP as a predictor for overall complications in abdominal surgery, IL-6 on day one predicted postoperative complications, whereas CRP did not.[10] Moreover, in our study, the confidence intervals for the c-statistic of IL-6 and CRP are wide and largely overlapping, indicating uncertainty in the predictive performance estimates. Therefore, subtle differences in predictive performance between IL-6 and CRP might have been concealed. Furthermore, physicians assess individual patient risk using a combination of clinical predictors, not on a single biomarker alone. The added value of a biomarker differs, relative to the other predictors used to assess infection risk. As the optimal set of predictors for postoperative infection risk assessment still has to be investigated, we cannot conclude yet on the superiority of either IL-6 or CRP when used in combination with such a set of predictors for postoperative infection. Given the timing of peak concentrations of IL-6 relative to CRP, there might be an advantage in infection risk prediction for IL-6 when assessing infection risk directly postoperative. CRP, on the other hand, might have a logistical advantage on the first postoperative morning, as CRP concentrations gradually increase, reducing the relevance of sample timing to detect the maximum concentration.

Nowadays postoperative care in lung surgery has put focus on enhanced recovery after surgery (ERAS) programs resulting in optimized pain management, direct mobilization after surgery and early drain removal. This has resulted in early recovery and hospital discharge [17]. Supposedly, a future risk prediction model should align with these developments, further optimizing its use.

This study has several strengths. First of all, we used a multivariable approach to evaluate the predictive performance of the inflammatory biomarkers. Secondly, we used a clinically relevant prediction horizon, leaving enough time for preventive treatment between infection prediction and diagnosis. Thirdly, we used clear infection outcome definition according to widely accepted international criteria, in combination with prospectively collected biomarker- and clinical data. There are, however, also several limitations. We evaluated the added value of the inflammatory biomarkers relative to a core model based on clinical reasoning. Ideally, we would have investigated incremental value on a validated postoperative infection risk model in pulmonary surgery. Secondly, the limited sample size resulted in broad confidence intervals of the c-statistic and other performance measures. As a consequence, a moderate incremental value may have been missed. Additionally, the sample size was too small to analyse independent predictive performance relative to open vs. minimally invasive surgery, which is known to affect postoperative inflammatory biomarker concentrations [10]. Finally, the selected time points for blood sampling may not coincide with the exact peak concentration of IL-6. Taken together, we cannot conclude yet on the clinical relevance of IL-6 and CRP demonstrated by these predictive performance measures.

## Conclusion

In conclusion, early postoperative plasma IL-6 and CRP concentrations are independently associated with subsequent infection risk, although neither biomarker improved prognostic classification by a simple prediction model using readily available clinical data. However, our study was small and limited to pulmonary surgery only. Therefore further research investigating the prognostic value of IL-6 and CRP relative to other predictors of postoperative infection is needed to draw final conclusions on the clinical utility of either biomarker.

## Supporting information

**S1 File.  Supporting information.** Patient enrolment, median perioperative biomarker concentrations, infection definitions, and univariable logistic regression analyses.
(DOCX)

## Author contributions

**Conceptualization:** Ted Reniers, Peter G. Noordzij, Eelco J. Veen, Anne Marlies Taselaar, Thijs C.D. Rettig.

**Formal analysis:** Ted Reniers, Peter G. Noordzij, Lisette M. Vernooij.

**Funding acquisition:** Thijs C.D. Rettig.

**Investigation:** Ted Reniers, Peter G. Noordzij, Eelco J. Veen, Erik F.N. Hofman, Anne Marlies Taselaar, W. Anton Visser, Pim van der Heiden, Stefan Boeckx, Thijs C.D. Rettig.

**Methodology:** Ted Reniers, Peter G. Noordzij, Olaf L. Cremer, Lisette M. Vernooij, Thijs C.D. Rettig.

**Project administration:** Peter G. Noordzij, Eelco J. Veen, Thijs C.D. Rettig.

**Resources:** Peter G. Noordzij, Eelco J. Veen, Judith M.A. Emmen, Ineke M. Dijkstra, Thijs C.D. Rettig.

**Supervision:** Peter G. Noordzij, Lisette M. Vernooij, Thijs C.D. Rettig.

**Visualization:** Ted Reniers.

**Writing – original draft:** Ted Reniers, Peter G. Noordzij, Thijs C.D. Rettig.

**Writing – review & editing:** Ted Reniers, Peter G. Noordzij, Eelco J. Veen, Erik F.N. Hofman, Anne Marlies Taselaar, W. Anton Visser, Pim van der Heiden, Stefan Boeckx, Judith M.A. Emmen, Ineke M. Dijkstra, Olaf L. Cremer, Lisette M. Vernooij.

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
