## [Decision Letter · Decision Letter 0]

PONE-D-25-06586Does postoperative IL-6 improve early prediction of infection after pulmonary cancer surgery? A two-centre prospective studyPLOS ONE

Dear Dr. Reniers,

Thank you for submitting your manuscript to PLOS ONE. After careful consideration, we feel that it has merit but does not fully meet PLOS ONE’s publication criteria as it currently stands. Therefore, we invite you to submit a revised version of the manuscript that addresses the points raised during the review process.

We look forward to receiving your revised manuscript.

Kind regards,

Ennio Polilli

Academic Editor

PLOS ONE

Journal Requirements:

“P.G. Noordzij is a member of the advisory board of Roche Diagnostics on the perioperative use of biomarkers. P.G. Noordzij, T.C.D. Rettig and T. Reniers conduct a separate research study on perioperative biomarkers funded by Roche Diagnostics. All other authors have no conflicts of interest.”

4. Please include captions for your Supporting Information files at the end of your manuscript, and update any in-text citations to match accordingly. Please see our Supporting Information guidelines for more information: http://journals.plos.org/plosone/s/supporting-information .

Reviewers' comments:

Reviewer's Responses to Questions

**Comments to the Author**

1. Is the manuscript technically sound, and do the data support the conclusions?

Reviewer #1: Yes

Reviewer #2: Partly

Reviewer #3: Yes

Reviewer #4: Partly

2. Has the statistical analysis been performed appropriately and rigorously? 

Reviewer #1: Yes

Reviewer #2: Yes

Reviewer #3: I Don't Know

Reviewer #4: Yes

3. Have the authors made all data underlying the findings in their manuscript fully available?

Reviewer #1: Yes

Reviewer #2: Yes

Reviewer #3: Yes

Reviewer #4: Yes

4. Is the manuscript presented in an intelligible fashion and written in standard English?

Reviewer #1: No

Reviewer #2: Yes

Reviewer #3: Yes

Reviewer #4: Yes

5. Review Comments to the Author

Reviewer #1: Dear Authors

You have done a fascinating research, but there are some issues that should be solved:

1. Key words must be reconsidered.

2. title is not attracting enough. for this high level kind of work, there must be a better title.

3. the language and the grammar is not acceptable for publication.

4. your references are not up to dated. please provide the latest related references.

5. introduction part has not introduced the Problem.

6. most importantly, the discussion part. you should compare your work to the latest related research's and write what makes your research Distinctive.

7.. you said " The study was preregistered on the 13th

of July 2018, prior to conducting the research with The Netherlands Trial Register (NTR), currently

available via the International Clinical Trial Registry Platform (ICTRP), with registration number NLOMON25075.

Ethical approval of the study protocol was obtained on the 23th of July 2018 by the

‘Toetsingscommissie Wetenschappelijk Onderzoek Rotterdam e.o.’ (TWOR, number NL.64754.101.18)". what took you research so long to be completed? just a question out of curiosity.

Kind regards

Reviewer #2: The study could not elucidate a conclusive findings. The !L-6 biomarker has been hypothized to be associated with post-operative infection which has not been established in this study.

It is concluded in the manuscript that "There are, however, also several limitations. We evaluated the added value of the inflammatory biomarkers relative to a core model based

on clinical reasoning. Ideally we would have investigated incremental value on a validated postoperative

infection risk model in pulmonary surgery. Secondly, the limited sample size resulted in broad confidence

intervals of the c-statistic and other performance measures. As a consequence, a moderate incremental

value may have been missed. Additionally, the sample size was too small to analyse independent

predictive performance relative to open vs. minimal invasive surgery, which is known to affect

postoperative inflammatory biomarker concentrations.(9) Taken together, we cannot conclude yet on the

clinical relevance of IL-6 and CRP demonstrated by these predictive performance measures."

Reviewer #3: Thank you for conducting this interesting study. Here are my comments that help improve the paper.

Short title: I suggest the short title to be “IL-6 to predict infection in pulmonary surgery”

Introduction: It would be better if more detailed information is added on sepsis and inflammation.

Results: The first paragraph under study population section in your result shows the process to reach your sample size and should be in methods and materials section under inclusion & exclusion criteria. Under postoperative infection section, what is the difference between the 28 postoperative infections and the 45 postoperative infection events? Either it should be rewritten or amended based on the actual findings. Please clarify it. In biomarker kinetics section, the concentrations of biomarkers (IL-6, CRP) before and after surgery/operation needs to be clarified. For example, the concentration of IL-6 is 6.1pg/ml (preoperative) and 3.4pg/ml (postoperative); can it be? Clarify.

Introduction: In the result you stated that IL-6 was associated with postoperative infection. However, your discussion/conclusion didn’t support this finding. What is your basis for this conclusion? Justify it.

Reviewer #4: Dear authors,

very good research and the findings add to already existing knowledge we have on the risk of postoperative infections.

Here are a few comments/questions for your attention.

Review: Does postoperative IL-6 improve early prediction of infection after pulmonary cancer

surgery? A two-centre prospective study

General Comments

- Overall some of the findings of this paper are very important. Understanding the added value of IL-6 and CRP levels as an early predictor of infection after pulmonary cancer surgery is essential.

- The knowledge is useful for patient care and aid clinicians to minimize the risk of postoperative infection. Very good attempt by the authors.

Section specific comments

Title

- It could be worded to Does postoperative serum or plasma IL-6 improve early prediction of infection after pulmonary cancer surgery? A two-centre prospective study

- In the manuscript reference is made to blood sample but it’s not clear if plasma or serum was used. Using IL-6 could be misleading, a more specific matrix linked to the IL-6 levels would be appropriate, e.g. serum IL-6 or Plasma IL-6

Summary/Abstract

- Please clarify in the conclusion whether IL-6 is plasma or serum

Introduction

- It is good that the authors mentioned the well-established fact that IL-6 concentrations are elevated after surgery, trauma, and clinical illness.

Methods

- Study procedures and biomarker sampling – the authors mention “Blood samples were drawn from an arterial line directly after induction of general anaesthesia (further referred to as ‘preoperative sample’), and after 6, 9, 12, 24, 48, and 72 hours. Blood samples were

centrifuged and stored in a freezer at -80 °C and shipped to the clinical chemistry lab of the Sint Antonius hospital Nieuwegein for batch analysis.”

a. Can the authors specify what type of blood collection tubes, how many tubes and the volume of blood drawn?

b. Is it correct to assume that EDTA-blood was used for WBC measurements? Authors should provide clarity in the text.

c. What types of matrix was derived from the blood samples that were centrifuged and stored in a freezer at -80 °C? plasma or serum?

d. The goal was to determine if IL-6 and CRP levels observed within the first 24 hours after surgery are associated with postoperative infection risk. What is the justification of not including timepoints 0 and 3 hours?

Results

- In Table 1, there is significant difference for the surgery duration (in minutes) and blood loss between patients with and without postoperative infection. Considering that patients with infection had higher IL-6 levels, did the authors find any correction between the surgery time/intraoperative blood loss with the IL-6 levels?

- Since preoperative (Timepoint 0) and 3 hour postoperative levels of IL-6 were not presented in this study, how do the authors justify that Postoperative peak IL-6 concentrations occurred after 6 hours in the majority of patients?

- Base on the results, can the authors base their findings to IL-6 levels possibly reflect the severity of postoperative infection?

Discussion

- The general presentation in the discussion section looks ok.

- Considering the weight of gender(male) and age to their predictive performance model, the arguments could have benefitted from it although those facts are well established.

- Authors could consider the delta change in these biomarkers respective to the timepoints as a marker of postoperative infection risk, severity and/or recovery outcomes.

Figure 1

Preoperative (Time=0 hours) levels of the biomarkers are shown. Throughout the manuscript, there is no mention of this timepoint sampling. Where did this timepoint come from and were the levels indeed zero?

6. PLOS authors have the option to publish the peer review history of their article (what does this mean? ). If published, this will include your full peer review and any attached files.

**Do you want your identity to be public for this peer review?** For information about this choice, including consent withdrawal, please see our Privacy Policy .

Reviewer #1: **Yes: ** Mohammad Amin Bakhshan

Reviewer #2: No

Reviewer #3: No

Reviewer #4: **Yes: ** Daniel Antwi-Berko

---

## [Author Response · Author response to Decision Letter 1]

14 May 2025

Response to the reviewers

Reviewer #1:

Dear Authors

You have done a fascinating research, but there are some issues that should be solved:

Dear reviewer,

Thank you for the positive feedback and for taking the time to carefully review our manuscript. We appreciate the constructive comments and will address the noted issues thoroughly in our revisions.

Kind regard,

Ted Reniers

Comment 1:

Key words must be reconsidered.

Answer 1:

We reconsidered our keywords and added “inflammation” and “prediction” to the existing keywords “pulmonary surgery”, “IL-6” and “infection”.

Comment 2:

title is not attracting enough. for this high level kind of work, there must be a better title.

Answer 2:

Thank you for your feedback regarding the title. We absolutely agree that an engaging title is important. However, we feel that the current title accurately reflects the content and scope of the work. We presented the title as a question to attract readers and intended to provide the right balance between clarity and relevance. To further improve clarity, we added a specification of “plasma” to the title.

Comment 3:

the language and the grammar is not acceptable for publication.

Answer 3:

We have carefully reviewed the manuscript for language and grammatical issues and made the necessary corrections to ensure it meets the standards required for publication.

Comment 4:

your references are not up to dated. please provide the latest related references.

Answer 4:

Thank you for your comment. We have reviewed and updated the references where appropriate.

Added references:

3. Andalib A, Ramana-Kumar A V, Bartlett G, Franco EL, Ferri LE. Influence of postoperative infectious complications on long-term survival of lung cancer patients: a population-based cohort study. J Thorac Oncol Off Publ Int Assoc Study Lung Cancer. 2013 May;8(5):554–61.

5. Prescott HC, Angus DC. Enhancing Recovery From Sepsis: A Review. JAMA. 2018 Jan;319(1):62–75.

16. Adamina M, Steffen T, Tarantino I, Beutner U, Schmied BM, Warschkow R. Meta-analysis of the predictive value of C-reactive protein for infectious complications in abdominal surgery. Br J Surg. 2015 May;102(6):590–8.

Comment 5:

introduction part has not introduced the Problem.

Answer 5:

Thank you for this comment. While we believe the introduction outlines the problem, we agree that the text would benefit from emphasising hyperinflammation as the key predictor of interest earlier in the text.

In the introduction, we first highlight the importance of timely identification of postoperative infections. Early identification of patients at risk could improve the monitoring of infectious signs and symptoms, thereby facilitating prompt diagnosis and treatment. While postoperative surgical hyperinflammation—regardless of the presence of infection—has been associated with subsequent postoperative infections, it remains unclear whether this response can help identify at-risk patients within 24 hours after surgery in a clinically meaningful way. In this context, we also provide the rationale for using IL-6 as a biomarker of postoperative hyperinflammation. Therefore, our aim was to investigate whether postoperative hyperinflammation, as indicated by peak IL-6 concentrations, could serve as an early predictor of postoperative infection.

For clarification we added the following sentence to the second paragraph of the introduction:

“Postoperative hyperinflammation might be the predictor to identify these patients at risk for infection.”

Comment 6:

most importantly, the discussion part. you should compare your work to the latest related research's and write what makes your research Distinctive.

Answer 6:

We appreciate this comment. We agree that the comparison with previous research on the topic is an important part of the discussion. The strictly predictive, multivariable approach focusing on IL-6 to predict infection is, to the best of our knowledge, lacking. However, there is a study on IL-6 as a predictor for severe postoperative complications. We added a section on this previous research in the second paragraph of the discussion. Additionally, we added some clarification of the third paragraph of the discussion. In this paragraph we explain what makes our research distinctive from other studies investigating inflammatory biomarkers in perioperative care. In paragraph four we further extend on the comparison with previous prediction research on IL-6 and CRP in abdominal surgery.

“Our study builds upon previous research in which the authors concluded univariable peak plasma IL-6 concentrations were associated with severe postoperative complications after pulmonary surgery.(11) We observed similar results regarding infection, although further exploration of predictive performance measures in a multivariable approach suggested limited clinical relevance of IL-6.

Several other studies investigated the association between inflammatory biomarkers and postoperative infection in various surgery types, however, most of these studies used an univariable, diagnostic approach.(9,16)” Discussion, page 15

Comment 7:

you said " The study was preregistered on the 13th of July 2018, prior to conducting the research with The Netherlands Trial Register (NTR), currently available via the International Clinical Trial Registry Platform (ICTRP), with registration number NLOMON25075. Ethical approval of the study protocol was obtained on the 23th of July 2018 by the ‘Toetsingscommissie Wetenschappelijk Onderzoek Rotterdam e.o.’ (TWOR, number NL.64754.101.18)".

what took you research so long to be completed? just a question out of curiosity.

Answer 7:

There are two important reasons for the delay in study completion. During the COVID-19 pandemic, study inclusion was lagging. Additionally, rapid recovery pathways after pulmonary surgery were introduced during the study period. Therefore, patients were increasingly discharged to the general ward and, in patients that were admitted to the ICU, arterial lines were removed as soon as possible. Considering the frequency of blood sampling in the study design, the burden of blood sampling without an arterial line would be too high and conflicting with the study procedures as approved by the medical ethics committee.  

Reviewer #2:

The study could not elucidate a conclusive findings. The !L-6 biomarker has been hypothized to be associated with post-operative infection which has not been established in this study.

It is concluded in the manuscript that "There are, however, also several limitations. We evaluated the added value of the inflammatory biomarkers relative to a core model based

on clinical reasoning. Ideally we would have investigated incremental value on a validated postoperative infection risk model in pulmonary surgery. Secondly, the limited sample size resulted in broad confidence intervals of the c-statistic and other performance measures. As a consequence, a moderate incremental value may have been missed. Additionally, the sample size was too small to analyse independent predictive performance relative to open vs. minimal invasive surgery, which is known to affect postoperative inflammatory biomarker concentrations.(9) Taken together, we cannot conclude yet on the clinical relevance of IL-6 and CRP demonstrated by these predictive performance measures."

Dear reviewer,

We sincerely appreciate your time and effort in evaluating our manuscript. While we acknowledge that our study has certain limitations, we believe it meaningfully contributes to the existing body of evidence by offering new insights into the predictive performance of IL-6 and other inflammatory biomarkers, particularly in comparison to readily available clinical predictors.

Although previous studies have suggested an association between these biomarkers and postoperative infections or complications, few have examined their added clinical relevance in prediction models. Our findings confirm that IL-6 and CRP are indeed associated with postoperative infection; however, our results also indicate that their incremental clinical utility remains moderate at most. We believe our study improves the understanding of the clinical relevance of inflammatory biomarkers to predict postoperative infection in a multivariable approach.

Kind regards,

Ted Reniers

Reviewer #3:

Thank you for conducting this interesting study. Here are my comments that help improve the paper.

Dear reviewer,

We sincerely thank you for your thoughtful and constructive feedback. We appreciate the time and effort taken to review our manuscript and are grateful for the valuable comments, which have helped us to improve the clarity and quality of the paper.

Kind regards,

Ted Reniers

Comment 1:

Short title: I suggest the short title to be “IL-6 to predict infection in pulmonary surgery”

Answer 1:

We added the short title as suggested.

Comment 2:

Introduction: It would be better if more detailed information is added on sepsis and inflammation.

Answer 2:

Thank you for your comment. We are not entirely certain about the specific type of information on sepsis and inflammation that you are referring to in the context of our manuscript’s aim. We attempt to explain our interpretation of the comment below:

We appreciate the importance of sepsis and inflammation. Although sepsis is very relevant regarding the rationale for infection prediction, the outcome we aim to predict is infection, regardless of sepsis. To maintain clarity and avoid redundancy, we have aimed to present a concise overview relevant to our objectives. Therefore, we believe that the information currently provided offers sufficient context for the scope and focus of our study.

We hope you find this approach appropriate.

Comment 3:

Results: The first paragraph under study population section in your result shows the process to reach your sample size and should be in methods and materials section under inclusion & exclusion criteria.

Answer 3:

We appreciate this comment. While we understand the concern raised, we need to adhere to the TRIPOD checklist. As outlined in this guideline, in the methods section the eligibility criteria should be presented and the results section should describe the “flow of participants through the study.” Accordingly, we have presented the inclusion and exclusion criteria in the “Design and Participants” section of the Methods and the first paragraph of the results section describes the number of patients that met these criteria.

We hope this explanation adequately addresses your concern.

Comment 4:

Under postoperative infection section, what is the difference between the 28 postoperative infections and the 45 postoperative infection events? Either it should be rewritten or amended based on the actual findings. Please clarify it.

Answer 4:

Some patients developed more than one infection event. We added a clarification within the text.

“In total, 38 (22%) patients developed a postoperative infection, with a total of 45 infection events (i.e. some patients developed more than one infection event).” Page 10, results paragraph 3

Comment 5:

In biomarker kinetics section, the concentrations of biomarkers (IL-6, CRP) before and after surgery/operation needs to be clarified. For example, the concentration of IL-6 is 6.1pg/ml (preoperative) and 3.4pg/ml (postoperative); can it be? Clarify.

Answer 5:

Thank you for this comment. The concentrations 6.1 and 3.4 represent the median preoperative concentrations in patients with postoperative infection vs. patients without postoperative infection. The postoperative concentrations are presented later on in this paragraph (page 13), again comparing patients with and without postoperative infection. We added clarification on the comparison made in the text.

“Preoperative median IL-6 and CRP concentrations were higher in patients with postoperative infection compared to non-infected patients (6.1 pg/ml (IQR 3.3,10.9) vs. 3.4 pg/ml (2.0,6.0), p=0.002 and 3.4 mg/L (2.1,22.1) vs. 1.8 mg/L (0.8,6.0), p=0.016 for IL-6 and CRP respectively, Figs 1A-B).” Results paragraph 4, page 12.

Comment 6:

In the result you stated that IL-6 was associated with postoperative infection. However, your discussion/conclusion didn’t support this finding. What is your basis for this conclusion? Justify it.

Answer 6:

We agree that the rationale underlying our conclusions could be described more clearly. Our study is prognostic and not etiologic. The goal is to assess the capability of IL-6 and other inflammatory biomarkers to estimate the probability of infection. We do not aim to investigate a causal relation between IL-6 and infection. To assess clinical relevance of the estimated outcome probability, measures such as discrimination, calibration and reclassification are more meaningful compared to association alone. doi: https://doi.org/10.1136/bmj.b375 Therefore, we based our conclusion on these measures. We added some information to clarify this statement in the first paragraph of the discussion: “Based on the predictive performance measures discrimination, calibration and reclassification, the added value of IL-6 and CRP to a simple clinical prediction model seems limited.”

Reviewer #4:

Dear authors, very good research and the findings add to already existing knowledge we have on the risk of postoperative infections. Here are a few comments/questions for your attention.

General Comments

- Overall some of the findings of this paper are very important. Understanding the added value of IL-6 and CRP levels as an early predictor of infection after pulmonary cancer surgery is essential.

- The knowledge is useful for patient care and aid clinicians to minimize the risk of postoperative infection. Very good attempt by the authors.

Dear reviewer,

We sincerely thank you for the kind and encouraging feedback. We are pleased to hear that the findings are considered valuable and relevant to clinical practice. We are grateful for the thoughtful comments and questions provided to help us further improve the manuscript.

Kind regards,

Ted Reniers

Section specific comments

Comment 1:

It could be worded to Does postoperative serum or plasma IL-6 improve early prediction of infection after pulmonary cancer surgery? A two-centre prospective study

Answer 1:

We changed the title as suggested.

Comment 2:

In the manuscript reference is made to blood sample but it’s not clear if plasma or serum was used. Using IL-6 could be misleading, a more specific matrix linked to the IL-6 levels would be appropriate, e.g. serum IL-6 or Plasma IL-6

Answer 2:

We agree this is important information. We added information on the matrix to the methods section.

“Measurements of plasma IL-6, CRP, and PCT were performed on an automated Cobas® 8000 platform (Roche Diagnostics, Mannheim, Germany).” Materials and methods paragraph 4, page 6.

Comment 3:

Please clarify in the conclusion whether IL-6 is plasma or serum

Answer 3:

We changed the conclusion to plasma IL-6 as suggested.

Comment 4:

It is good that the authors mentioned the well-established fact that IL-6 concentrations are elevated after surgery, trauma, and clinical illness.

Comment 5:

Study procedures and biomarker sampling – the authors mention “Blood samples were drawn from an arterial line directly after induction of general anaesthesia (further referred to as ‘preoperative sample’), and after 6, 9, 12, 24, 48, and 72 hours. Blood samples were centrifuged and stored in a freezer at -80 °C and shipped to the clinical chemistry lab of the Sint Antonius hospital Nieuwegein for batch analysis.”

a. Can the authors specify what type of blood collection tubes, how many tubes and the volume of blood drawn?

b. Is it correct to assume that EDTA-blood was used for WBC measurements? Authors should provide clarity in the text.

c. What types of matrix was derived from the blood samples that were centrifuged and stored in a freezer at -80 °C? plasma or serum?

Answer 5a-c:

Thank you. We added the following information to the methods section:

“Blood samples were drawn from an arterial line directly after induction of general anaesthesia (further referred to as ‘preoperative sample’), and af

---

## [Decision Letter · Decision Letter 1]

Does postoperative plasma IL-6 improve early prediction of infection after pulmonary cancer surgery? A two-centre prospective study

PONE-D-25-06586R1

Dear Dr. Reniers,

We’re pleased to inform you that your manuscript has been judged scientifically suitable for publication and will be formally accepted for publication once it meets all outstanding technical requirements.

Kind regards,

Ennio Polilli

Academic Editor

PLOS ONE

Additional Editor Comments (optional):

Reviewers' comments:

Reviewer's Responses to Questions

**Comments to the Author**

1. If the authors have adequately addressed your comments raised in a previous round of review and you feel that this manuscript is now acceptable for publication, you may indicate that here to bypass the “Comments to the Author” section, enter your conflict of interest statement in the “Confidential to Editor” section, and submit your "Accept" recommendation.

Reviewer #1: All comments have been addressed

Reviewer #2: All comments have been addressed

Reviewer #4: All comments have been addressed

2. Is the manuscript technically sound, and do the data support the conclusions?

Reviewer #1: Yes

Reviewer #2: Yes

Reviewer #4: Yes

3. Has the statistical analysis been performed appropriately and rigorously? 

Reviewer #1: Yes

Reviewer #2: Yes

Reviewer #4: Yes

4. Have the authors made all data underlying the findings in their manuscript fully available?

Reviewer #1: Yes

Reviewer #2: Yes

Reviewer #4: Yes

5. Is the manuscript presented in an intelligible fashion and written in standard English?

Reviewer #1: Yes

Reviewer #2: Yes

Reviewer #4: Yes

6. Review Comments to the Author

Reviewer #1: (No Response)

Reviewer #2: Manuscript has been improved

Still the title can be improved further to become more meaningful

The conclusion should be more clear and precise. Concrete findings would be more helpful for further studies...

Reviewer #4: (No Response)

7. PLOS authors have the option to publish the peer review history of their article (what does this mean? ). If published, this will include your full peer review and any attached files.

**Do you want your identity to be public for this peer review?** For information about this choice, including consent withdrawal, please see our Privacy Policy .

Reviewer #1: **Yes: ** Mohammad Amin Bakhshan

Reviewer #2: No

Reviewer #4: **Yes: ** Daniel Antwi-Berko

---

## [Editor Report · Acceptance letter]

PONE-D-25-06586R1

PLOS ONE

Dear Dr. Reniers,

I'm pleased to inform you that your manuscript has been deemed suitable for publication in PLOS ONE. Congratulations! Your manuscript is now being handed over to our production team.

Kind regards,

on behalf of

Dr. Ennio Polilli

Academic Editor

PLOS ONE